# Tensile Performance Test Research of Hybrid Steel Fiber—Reinforced Self-Compacting Concrete

**DOI:** 10.3390/ma16031114

**Published:** 2023-01-27

**Authors:** Chenjie Gong, Lei Kang, Wenhan Zhou, Linghui Liu, Mingfeng Lei

**Affiliations:** School of Civil Engineering, Central South University, Changsha 410075, China

**Keywords:** hybrid steel fiber–reinforced self-compacting concrete, tensile properties, DIC technology, fiber content

## Abstract

Notched beam specimens were loaded by the three-point bending test device, and the effects of different volume contents and combinations of steel fibers on the tensile properties of hybrid steel fiber–reinforced self-compacting concrete (HSFRSCC) were studied. The failure law and strain field distribution of the specimens were studied by digital image correlation (DIC) technology. Moreover, the curves between the load and crack mouth opening displacement (CMOD) of 18 groups of hybrid steel fiber–reinforced concrete specimens were obtained, and the stress–strain curves of 18 groups of specimens were derived from the load–CMOD curves. The results show that both single and hybrid steel fibers can improve the crack deformation resistance and tensile properties of concrete, but hybrid steel fibers have a more significant improvement effect. Only when the content of steel fiber is more than 0.6% can it have a more obvious postpeak descending section, and hybrid steel fiber has higher postpeak deformation capacity and flexural toughness. The fundamental reason why concrete with hybrid steel fibers has better tensile properties is that micro and macro steel fibers cooperate with each other to resist cracks, improving the toughness of concrete after cracking. Finally, the mechanism of different size and volume content of steel fiber was analyzed from the micro level, which can be used as a reference for the engineering design of HSFRSCC in the future.

## 1. Introduction

Concrete is the most widely used building material in the world, but ordinary concrete has some defects, such as low tensile strength, low toughness and easy cracking [1,2,3,4,5]. The addition of fiber can compensate for the above defects of concrete and improve the tensile strength, crack resistance and durability of concrete [6,7]. Steel fiber, as a common and economical inorganic fiber, is extensively used in the production of fiber-reinforced concrete (FRC) because of its high stiffness and strength, good adhesion property and relatively low cost. Moreover, steel fiber has the potential to replace part of conventional steel reinforcement and reduce the amount of steel reinforcement in concrete structure, so as to reduce self-weight and save cost [8,9,10,11]. To date, the experimental researches on the mechanical properties of steel fiber–reinforced concrete (SFRC) with mix ratio, steel fiber content and size as control variables have been abundant, and some of these research results are as follows.

Li et al. [12] studied the failure mode, load–displacement curve and ductility of concrete-filled steel tube columns and found that the addition of steel fiber can enhance the compressive strength and slightly improve the postpeak behavior and ductility of the specimens. Shi et al. [13] found that the addition of steel fiber can greatly improve the tensile performance of concrete, and the bridging effect of steel fiber can avoid the sudden loss of bearing capacity of plain concrete after reaching the tensile peak. Through the direct tensile test and DIC damage evaluation, Donnini et al. [14] observed that the addition of steel fiber can improve the tensile strength of concrete, and a high amount of steel fiber can obviously improve the initial crack tensile strength of the matrix. Abbass et al. [15] concluded that the length–diameter ratio of steel fiber has a certain effect on the direct tensile strength of concrete, and the steel fiber with a higher length–diameter ratio has a more obvious improvement on the direct tensile strength of concrete. Zhang et al. [16] analyzed the effects of fiber volume content and reinforcement ratio on the tensile properties of the concrete matrix and found that the strain hardening phenomenon of concrete with a higher content of steel fiber is more obvious.

The mechanical properties of steel fiber–reinforced concrete have been rich in experimental research results. It is concluded from the mechanical properties tests that under the same pouring quality, the addition of steel fiber can improve the compressive properties of concrete, but the improvement effect is not obvious, and the compressive strength is affected more by aggregate, pouring conditions and other factors. Moreover, when the content of steel fibers reaches a certain amount, excessive steel fibers tend to agglomerate in the concrete matrix, which is not conducive to the improvement of concrete compressive performance, and even reduces the strength of concrete [17,18,19]. The positive effect of steel fiber is mainly reflected in improving the tensile property, toughness and durability of concrete. Steel fiber in concrete can prevent the generation of new micro cracks, inhibit the through development of cracks, and thus improve the crack resistance of concrete. Meanwhile, there is an optimal mixing amount of steel fibers, which is greatly affected by fiber specification and mix ratio of concrete.

However, when the addition of a single type of fiber increases the mechanical properties of concrete on a certain scale, there are limitations of mechanical properties on another scale. For example, the deformed steel fiber has better drawing performance than straight steel fiber, while the high embedded friction can easily lead to matrix cracking, resulting in rapid crack propagation and reducing the ductility [20]. The microfiber can transfer the stress at the crack to the whole concrete matrix, improving the overall crack distribution mode. However, due to the lack of friction resistance, it is difficult to transfer the stress to the fiber; thus, the initial crack resistance performance is not obvious. Therefore, a single type of fiber has certain limitations on the improvement of concrete performance [21], which cannot meet the structural design and preparation requirements of high-performance concrete.

In recent years, many researchers have begun to study the effect of hybrid fibers on the comprehensive mechanical properties of concrete. Hybrid fibers can be divided in the following ways. One is the mixing of fibers of different materials, such as the mixing of polypropylene fiber and steel fiber, the mixing of steel fiber and polyvinyl alcohol (PVA) fiber, and the mixing of steel fiber and polyolefin fiber, etc. [22]. The other is the mixing of fibers of different sizes or shapes with the same material, such as the mixing of two polypropylene fibers of different lengths or the mixing of two steel fibers of different specifications. This kind of hybrid fiber mainly uses the different characteristics of one fiber to maximize the improvement of certain performances of concrete [23]. Some of the research results about hybrid fiber–reinforced concrete are as follows.

Chang et al. [24] conducted tests on the compressive and flexural mechanical properties of new active powdery concrete with hybrid steel fibers and found that the concrete specimen with 0.8% volume content of macrofibers and 0.2% volume content of microfibers had the highest bearing capacity and bending toughness. Chun et al. [25] found that with the increase of small-size steel fiber content, the average flexural strength and pullout energy of ultra-high-performance concrete (UHPC) with medium-size straight fibers have been increased. Yoo et al. [26] studied the hybrid effect of three kinds of steel fibers (straight fiber, end-hooked fiber and twisted fiber) in concrete and found that the concrete has the best flexural performance when 0.5% volume content of end-hooked steel fiber is mixed with 1.5% volume content of twisted steel fiber. Wu et al. [27] studied the mechanical properties of UHPC under static and dynamic loads by adding steel fibers with lengths of 6 mm and 13 mm, respectively. The results show that the UHPC mixed with 1.5% volume content of long fibers and 0.5% volume content of short fibers has the best static and dynamic mechanical properties. Ngoc et al. [28] studied the tensile properties of ultra-high-performance concrete mixed with 30 mm and 19 mm smooth steel fibers. The results show that the best multi-size hybrid synergistic effect of steel fibers on the peak strain-bearing property and toughness of the concrete matrix can improve the tensile properties of the concrete matrix at high strain rates.

It can be observed from the literature review that the existing researches about hybrid steel fibers are mostly based on the hybrid steel fiber–reinforced ultra-high-performance concrete (HSFRUHPC) and seldom consider the effect of hybrid steel fibers on the mechanical properties of self-compacting concrete. Obviously, if the research results in the field of hybrid steel fiber–reinforced concrete were directly adopted for HSFRSCC, the applicability would be open to question. In addition, the working condition design of different size characteristics of steel fiber should be comprehensive enough to better analyze the influence of steel fiber size characteristics, content and hybrid degree on the tensile properties of steel fiber–reinforced self-compacting concrete. In this paper, in view of the deficiency of existing research, 18 groups of notched self-compacting concrete beam specimens with different single and hybrid steel fibers are designed, and the three-point bending tests of CMOD are performed. Based on the DIC technology, the effect of steel fiber size characteristics and content on the tensile properties of concrete is discussed. This study can provide experimental reference for the study of tensile properties of hybrid steel fiber–reinforced self-compacting concrete.

## 2. The Experimental Design

### 2.1. Experimental Materials

PO 52.5R ordinary Portland cement, high-quality I-grade fly ash and S105 slag powder were adopted in this study. A viscosity modifier and superplasticizer were used to adjust the segregation resistance and flow performance of concrete, respectively. The aggregates in the test are divided into fine sand and coarse gravel, and the particle classes are 0–5 mm and 5–10 mm, respectively. End-hooked steel fiber (HS) and straight steel fiber (SS) were selected to prepare HSFRSCC. The densities of the two types of steel fibers are 7800 kg/m^3^. The physical parameters of the steel fibers are summarized in Table 1, and the physical figure is shown in Figure 1.

To compensate for the defects in the workability caused by the aggregation of steel fibers and to form the favorable orientation of steel fibers, self-compacting concrete (SCC) was used as the concrete matrix material [29]. Through the previous concrete mix proportion test results, the mix proportion that meets the basic fluidity requirements was determined, as shown in Table 2. Based on the previous experiment against fiber volume, it is found that only when the total fiber volume of single steel fiber was limited to no more than 0.8% and the total fiber volume of hybrid steel fiber was not more than 1.2% can the workability of concrete be ensured. Therefore, the total volume of single steel fiber and hybrid steel fiber should not exceed 0.8% and 1.2%, respectively. In addition, the end-hooked steel fiber and straight steel fiber are set to four different volume fractions, which are 0.2%, 0.4%, 0.6% and 0.8%.

### 2.2. Tensile Strength Experimental Procedure

A total of 18 notched beams were designed and made in this experiment, and the mix design of concrete described in Table 2 is selected. The length of the specimen is 400 mm, the width is 100 mm, and the height is 100 mm, as shown in Figure 2. The specimens needed to wet saw a notch of width ≤ 5 mm and length of 20 mm at the center of the bottom surface, and the crack mouth opening displacement of the notch was measured by the extensometer at the bottom of the specimen [30]. The test was loaded by the WA-1000C electrohydraulic servo universal testing machine. The displacement was measured by the DWMY full-bridge strain displacement sensor, and the strain was recorded by the DM-YB1808 static resistance strain gauge and DIC technology. In this experiment, load control (the preloading process) was combined with displacement control (after that): preloading was carried out at a rate of 0.1 kN/s, and after the preloading reached 5 kN, the loading rate was controlled at 0.4 mm/min until the specimen was destroyed. The loading time, load, mid-span deflection and CMOD data were collected, and the load–CMOD curves were obtained. The setup of the tensile experiment is given in Figure 2.

### 2.3. DIC Experimental Procedure

This study uses DIC technology to describe the formation and development of cracks during loading [31]. DIC technology can visualize the overall strain of the material and obtain more complete strain data than strain gauges. The specific operations were as follows. First, the smooth and clean surface of the specimen was coated with white paint, and the white paint was sprayed with disordered black paint spots. Then, the high-pixel camera was set in the best position, ensuring good light conditions and making up the light when the natural light was insufficient. The image acquisition began after focusing on the acquisition area. Finally, the open-source two-dimensional image processing code was used to discretize the strain field in the whole process of specimen change.

## 3. Results and Analysis

### 3.1. Analysis of the Failure Process and DIC Results

The peak points and points with CMOD = 3.5 mm in the load–CMOD schematic curve are extracted as important reference points to characterize the tensile strength of steel fiber–reinforced concrete, as shown in Figure 3 and visualized analysis of the CMOD failure process and strain field was conducted. Since the failure of plain concrete is rapid and the characteristics of the CMOD curve are not obvious, only the DIC strain field distribution characteristics of the ultimate single steel fiber–reinforced concrete specimens (H_0_S_8_, H_8_S_0_) and ultimate hybrid steel fiber–reinforced concrete specimens (H_6_S_6_) are analyzed, as shown in Figure 4, Figure 5 and Figure 6.

Figure 4, Figure 5 and Figure 6 show that the main crack of the single end-hooked steel fiber–reinforced concrete specimen does not develop along the notch but is located at the upper right of the notch, indicating that random initial microcracks have possibly existed due to the difference in pouring and curing conditions before mechanical loading. In the peak-strain stage, the maximum strains of straight steel fiber–reinforced concrete, end-hooked steel fiber–reinforced concrete and hybrid steel fiber–reinforced concrete are 1.5 × 10^−3^, 8 × 10^−3^ and 0.01, respectively. It shows that hybrid steel fiber significantly improves the crack resistance property and anti-crack ductility of specimens in the failure process. The strain concentration area of hybrid steel fiber–reinforced concrete is the smallest and the degree of expansion is the lowest.

In addition, according to the observation of the fracture surface of the specimen, steel fibers are mostly distributed in the loading tangential direction and perpendicular to the propagation direction of the crack, as shown in Figure 7. The two ends of steel fiber have a good bonding effect in concrete, so that it can give full play to the bridging role against the existing cracks and inhibit the upward expansion of cracks. Hence, compared with the plain concrete, the failure of the CMOD bending specimen with steel fiber changes from brittle failure to ductile failure.

### 3.2. Analysis of the Load-CMOD Evolution Curves

Figure 8 records the load–CMOD curves obtained from the CMOD three-point bending test. Through the analysis of each curve, the following conclusions can be drawn. For plain concrete, after reaching the peak value, the curve falls rapidly, and the load drops rapidly to 0 kN, showing brittle failure. For single steel fiber specimens, the curve shape of 0.2% and 0.4% straight steel fiber–reinforced concrete is similar to that of plain concrete. Although the peak value increases slightly, brittle failure still occurs. When the fiber content reaches 0.6%, the decreasing section of the curve slows down, and the deformation ability begins to improve. The curves of end-hooked steel fiber–reinforced concrete gradually increase with increasing fiber content. When the fiber content reaches 0.8%, the curve rises significantly, and the peak value is close to 18 kN. However, the postpeak curve decreases quickly, showing little toughness.

For hybrid steel fiber–reinforced concrete, there is an obvious postpeak descending section in each curve of the specimen, and it rises gradually with increasing total fiber volume. The postpeak deformation capacity of hybrid steel fiber concrete is significantly improved, and a higher load can be borne with slow failure, showing good toughness. Obviously, the addition of hybrid steel fibers can significantly improve the CMOD tensile properties of concrete and enhance the postpeak tensile toughness of concrete. Moreover, the slope of the postpeak curve of the H_6_S_6_ specimen is the smoothest, which means that after the peak, the loading rate is the slowest for this case, and the peak value is more than 15 kN. Although the H_8_S_0_ specimen has a higher peak value, the H_6_S_6_ specimen has good bearing capacity and postpeak behavior simultaneously and is considered to be the best steel fiber combination.

### 3.3. Parameter Analysis of Tensile Property

In this study, the tensile properties of steel fiber–reinforced concrete after cracking are characterized by calculating the residual bending tensile strength of each specimen. The residual bending tensile strength is defined by CMOD, and the calculation formula is shown in Equations (1) and (2). The characteristic residual tensile strength includes the peak residual bending tensile strength *f_ctm_*, *f_l_* and the corresponding residual tensile strength *f_R,j_* when CMOD is 0.5 mm, 1.5 mm, 2.5 mm and 3.5 mm, respectively. The calculated property parameters of the CMOD three-point bending test are shown in Table 3,
(1)$$f_{ctm,fl}=\frac{3F_{L}l}{2bh_{sp}^{2}}$$
(2)$$f_{R,j}=\frac{3F_{j}l}{2bh_{sp}^{2}}$$
where *f_ctm,fl_* is the peak residual bending tensile strength (MPa); *F_L_* is the peak load of CMOD in the range of 0–0.05 mm (N); *l* is the span of the specimen (mm); *b* is the width of the specimen (mm); *h_sp_* is the distance between the tip of the notch and the top of the specimen (mm); *f_R,j_* is the residual bending tensile strength (MPa) corresponding to CMOD = CMOD_j_, and *F_j_* is the load corresponding to CMOD = CMOD_j_ (N), as shown in Figure 9.

Table 3 shows that the tensile strength of plain concrete is low, and after CMOD reaches 1.5 mm, the residual load is close to 0 kN. The residual tensile strength basically disappears, and the concrete is destroyed rapidly after cracking, showing brittle failure and no ductility. The end-hooked steel fiber–reinforced concrete with 0.8% content has a higher ultimate tensile capacity, which is 150% of that of plain concrete, and the residual tensile load after cracking is better. It shows a strong tensile capacity; 0.8% straight steel fiber can also improve the tensile capacity of concrete, and the improvement effect is 190% of that of plain concrete. Compared with single steel fibers, the effect of hybrid steel fibers on tensile strength is more obvious. The tensile strength of the ultimate mixed specimen H_6_S_6_ is the highest, up to 10.85 MPa, which is 240% of that of plain concrete. The tensile strength after cracking is higher as a whole and decreases slowly with increasing CMOD, indicating that the addition of hybrid steel fibers can significantly improve the tensile performance of concrete. Different sizes of steel fibers cooperate with anti-cracks in the concrete matrix to improve the postcrack toughness of concrete.

### 3.4. Analysis of the Stress–Strain Evolution Curves

According to the “RILEM TC 162TDF” standard [32], the load–CMOD curve obtained from the CMOD three-point bending test of hybrid steel fiber–reinforced concrete can be transformed into a tensile stress–strain constitutive curve by the three-point method, as shown in Figure 10. The specific derivation process of the transformation method is shown in Equations (3)–(9).
(3)$$\sigma_{1}=0.7f_{ctm,fl}(1.6-d)$$
(4)$$\varepsilon_{1}=\frac{\sigma_{1}}{E_{c}}$$
(5)$$\sigma_{2}=0.45f_{R,1}\kappa_{h}$$
(6)$$\varepsilon_{2}=\varepsilon_{1}+0.1‰$$
(7)$$\sigma_{3}=0.37f_{R,4}\kappa_{h}$$
(8)$$\varepsilon_{3}=25‰$$
(9)$$E_{c}=9500{\left(f_{cm}\right)}^{1/3}$$
where *σ*_i_ (i = 1, 2, 3) is the characteristic stress of the tensile stress–strain curve; *ε*_i_ (i = 1, 2, 3) is the characteristic strain of the tensile stress–strain curve; d is the thickness of the specimen; *E_c_* is the equivalent elastic modulus; *κ_h_* is the size factor, which is equal to 1 when h < 10 cm; *f_cm_* is the average compressive strength of the cylinder specimen; and the conversion coefficient of the cube compressive strength is 0.79.

Through the above derivation formula, suitable tensile stress–strain curves can be obtained by calculating the three pairs of characteristic stress and strain values of each specimen and connecting the three curve characteristic points of each group. The 18 groups of specimens in the test were also divided into five groups: straight steel fiber, end-hooked steel fiber, and 0.2%, 0.4% and 0.6% end-hooked steel fiber. The tensile stress–strain curves were drawn, as shown in Figure 11.

As shown in Figure 11, compared with plain concrete, the tensile stress–strain curves of concrete mixed with steel fiber are raised upward, indicating that steel fiber can better improve the tensile performance of concrete. The addition of straight steel fibers increases the peak stress (tensile strength) of the concrete matrix, and the peak stress increases with increasing steel fiber content up to 9 MPa, which is 200% of that of plain concrete, as shown in Figure 11a. The end-hooked steel fiber decreases slowly after the peak of the curve, showing better tensile toughness, as shown in Figure 11b.

The improvement of the hybrid steel fiber on the tensile properties has a synergistic effect. Taking 0.6% end-hooked steel fiber as an example, the peak stress increases significantly and the postpeak stress decreases slowly with increasing fiber content. The tensile stress of H_6_S_6_ is the highest, which is higher than that of the ultimate single steel fiber–reinforced concrete and 260% higher than that of plain concrete, indicating that different sizes of hybrid steel fiber play a synergistic role in the concrete matrix.

## 4. Discussion


(1)Concrete is composed of multiscale and multiphase composite materials such as coarse aggregate, sand and cement. In hardened concrete, the joint surface of different materials is the weakest, which will lead to initial microcracks in the concrete. Under the effect of an external force, these microcracks gradually form macroscopic cracks and connect with each other and finally form the fracture surface, which leads to the failure of concrete. With the addition of straight steel fiber or end-hook steel fiber, crack propagation can be significantly restrained, and the stress concentration at the crack can be avoided by the high tensile strength and good bridging effect of steel fiber [33]. Compared with plain concrete, the expansion of initial cracks and the generation of new cracks lag obviously, which is the fundamental reason why the tensile strength of steel fiber–reinforced concrete is higher than that of plain concrete.(2)Hybrid steel fibers can improve the tensile performance of concrete better than single steel fibers. From the test results, it can be found that H_6_S_6_ is the best proportion of hybrid steel fiber–reinforced concrete. This is because in the elastic and plastic stage of steel fiber–reinforced concrete, the densely distributed straight steel fiber occupies the main inhibitory effect on the expansion of microcracks, and the primary cracks develop independently and do not connect with each other under the action of straight steel fiber. In the peak stage and failure stage, straight steel fiber continues to play a role in crack resistance, but it is no longer enough to restrain the development of macroscopic cracks, and end-hook steel fiber begins to play a leading role. In addition, with the continued effect of external forces, although there have been macroscopic cracks in the concrete, the concrete can still continue to bear load until the concrete is damaged. The mechanism analysis of the tensile properties of hybrid steel fiber–reinforced concrete is shown in Figure 12. Obviously, different sizes of steel fibers can produce a good synergistic effect to restrain the development of cracks in various stages.(3)Because of the larger mass, the end-hook steel fiber will move more intensively compared with the straight steel fiber in the process of concrete pouring and forming, thus acting as a miniature mixing rod or vibrating rod, making the interior structure of the concrete more uniform and resulting in fewer microcracks [34]. For straight steel fiber, in view of the fact that the matrix mix of each specimen is the same, since straight steel fiber has a larger specific surface area and more quantity, strong adsorption of straight steel fiber to paste will occur. Hence, this leads to a sparser pore structure of the concrete matrix, resulting in more microcracks [35]. Moreover, the addition of steel fiber will actually cause a greater disturbance to the packing structure of the aggregate, thus increasing the voids between the aggregates, as shown in Figure 13. These voids require more paste to fill [36], which will also lead to the loosening void structure of the matrix. However, because steel fiber has good bridging and crack resistance, despite the above negative effect, steel fiber reinforced concrete will still show better tensile properties than plain concrete as a whole.


## 5. Conclusions


(1)Hybrid steel fibers can significantly improve the tensile properties of concrete. The steel fiber in the cross section of the crack fully plays the role of bridging, restrains the expansion and connection of the crack, and makes the specimen change from brittle failure to ductile failure. The tensile strength of the ultimate hybrid specimen H_6_S_6_ is high, which is 200% of that of plain concrete, and its tensile strength after cracking is higher and decreases slowly with increasing CMOD. Different sizes of steel fibers cooperate with the matrix to resist cracks.(2)The tensile stress–strain curves are derived from the load–CMOD curves. From the tensile constitutive curve, it is found that compared with the plain concrete, the tensile stress–strain curve of the concrete mixed with steel fiber rises upward. The peak tensile stress of specimen H6S6 with the ultimate hybrid fiber content is 260% of that of the plain concrete and is only lower than that of the H_8_S_0_ specimen, and the H6S6 specimen has better postpeak behavior, indicating that the best combination of steel fibers is H_6_S_6_.(3)Different sizes of steel fibers can produce a positive hybrid effect in concrete. The analysis of the effect mechanism of hybrid steel fiber shows that straight steel fiber plays a role in the development period of concrete microcracks, and end-hooked steel fiber plays a role in the period of macroscopic crack propagation. The two kinds of steel fiber cooperate to restrain the development of cracks. The tensile failure of concrete is transformed from brittleness to ductility, which significantly improves the tensile performance of concrete.


## Figures and Tables

**Figure 1 materials-16-01114-f001:**
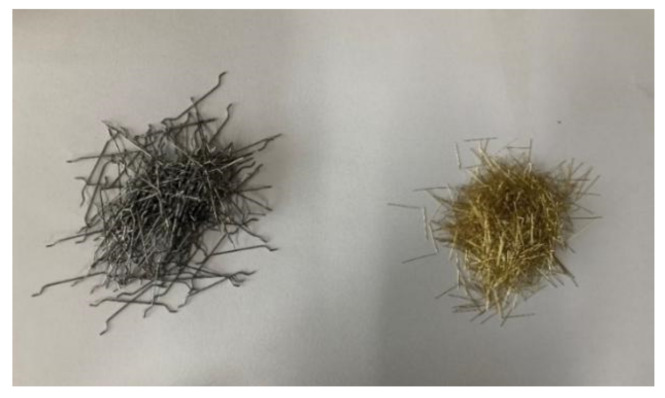
End-hooked steel fiber (left) and straight steel fiber (right).

**Figure 2 materials-16-01114-f002:**
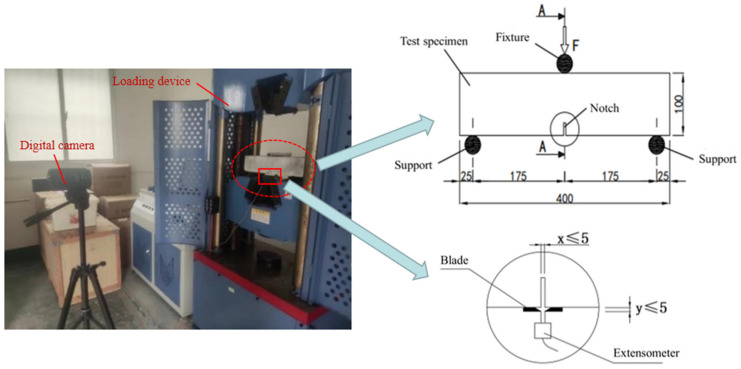
Setup figure of the CMOD test (unit: mm) [30].

**Figure 3 materials-16-01114-f003:**
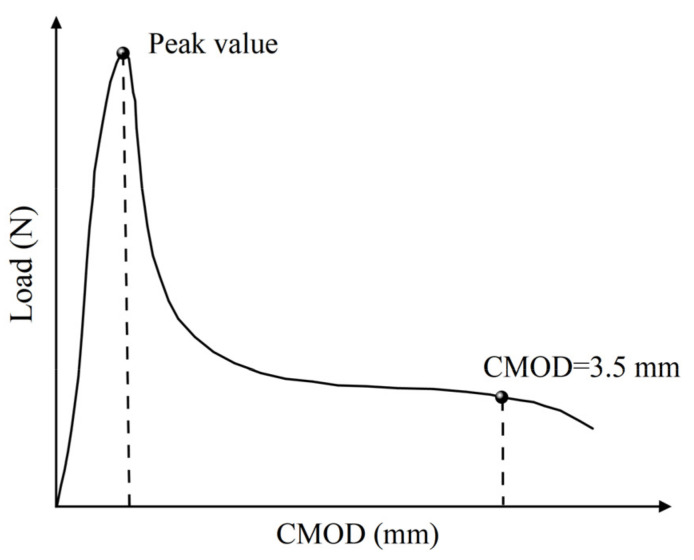
Load–CMOD schematic curve.

**Figure 4 materials-16-01114-f004:**
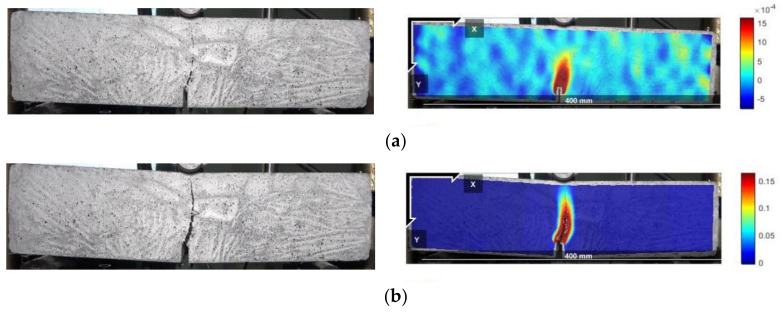
Failure process (left) and DIC strain field (right) of the H_0_S_8_ specimen. (**a**) Peak value; (**b**) CMOD = 3.5 mm.

**Figure 5 materials-16-01114-f005:**
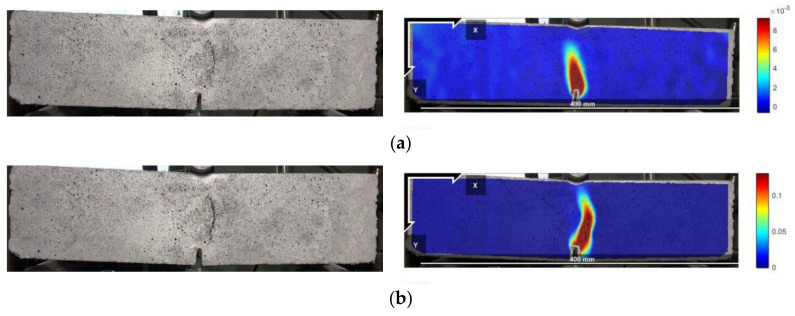
Failure process (left) and DIC strain field (right) of the H_8_S_0_ specimen. (**a**) Peak value; (**b**) CMOD = 3.5 mm.

**Figure 6 materials-16-01114-f006:**
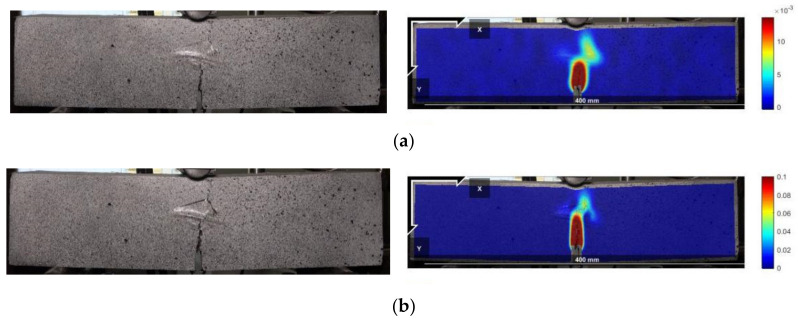
Failure process (left) and DIC strain field (right) of the H_6_S_6_ specimen. (**a**) Peak value; (**b**) CMOD = 3.5 mm.

**Figure 7 materials-16-01114-f007:**
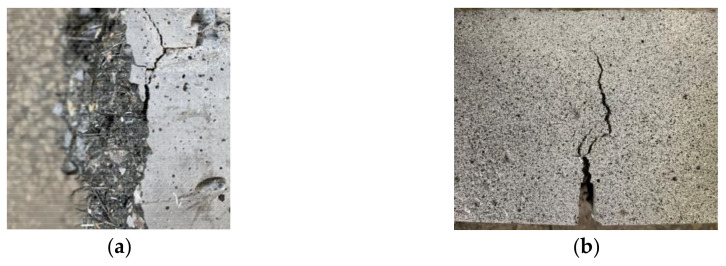
Steel fiber distribution in cross section and crack development of notch. (**a**) Steel fiber distribution in the cross section; (**b**) Crack development of notch.

**Figure 8 materials-16-01114-f008:**
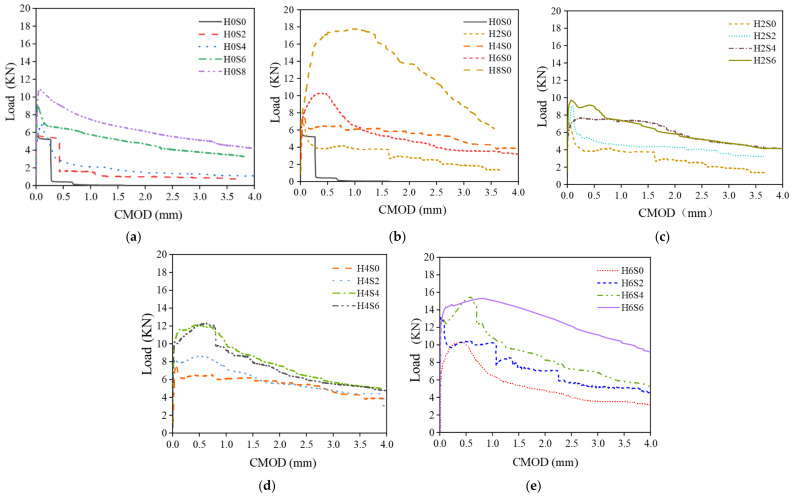
Load–CMOD experimental curve. (**a**) Straight fiber—single; (**b**) End-hooked fiber—single; (**c**) 0.2% end-hooked fiber—hybrid; (**d**) 0.4% end-hooked fiber—hybrid; (**e**) 0.6% end-hooked fiber—hybrid.

**Figure 9 materials-16-01114-f009:**
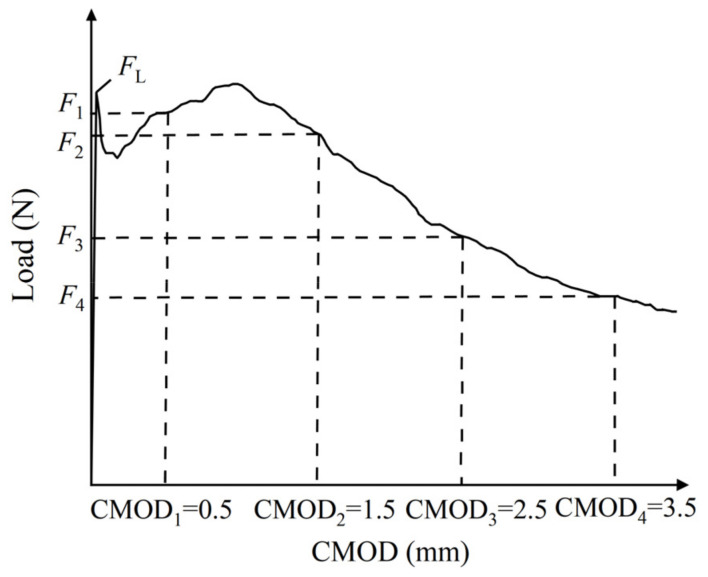
Load–CMOD characteristic curve [30].

**Figure 10 materials-16-01114-f010:**
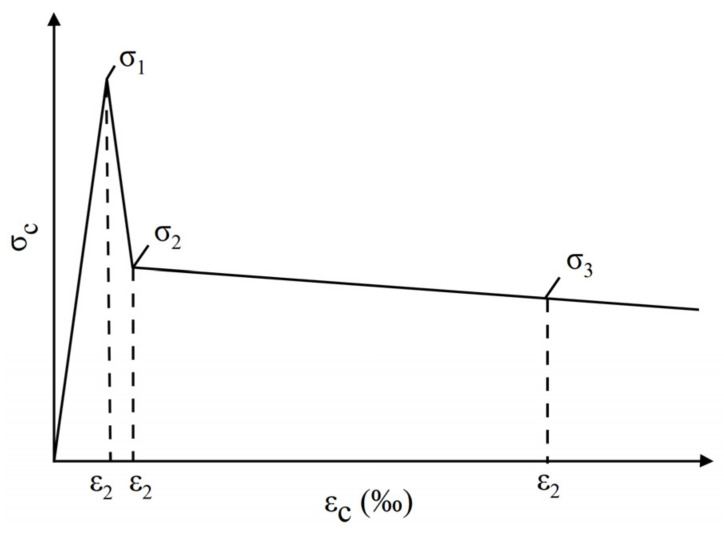
Tensile constitutive derivation curve [32].

**Figure 11 materials-16-01114-f011:**
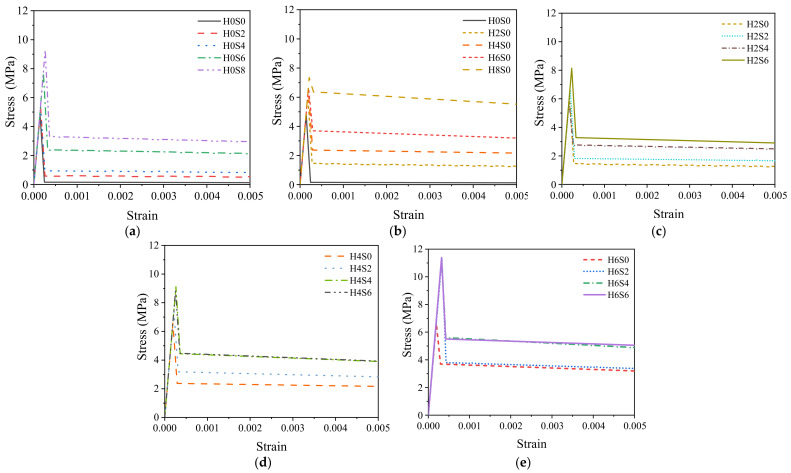
Tensile stress–strain curve. (**a**) Straight fiber—single; (**b**) End-hooked fiber—single; (**c**) 0.2% end-hooked fiber—hybrid; (**d**) 0.4% end-hooked fiber—hybrid; (**e**) 0.6% end-hooked fiber—hybrid.

**Figure 12 materials-16-01114-f012:**
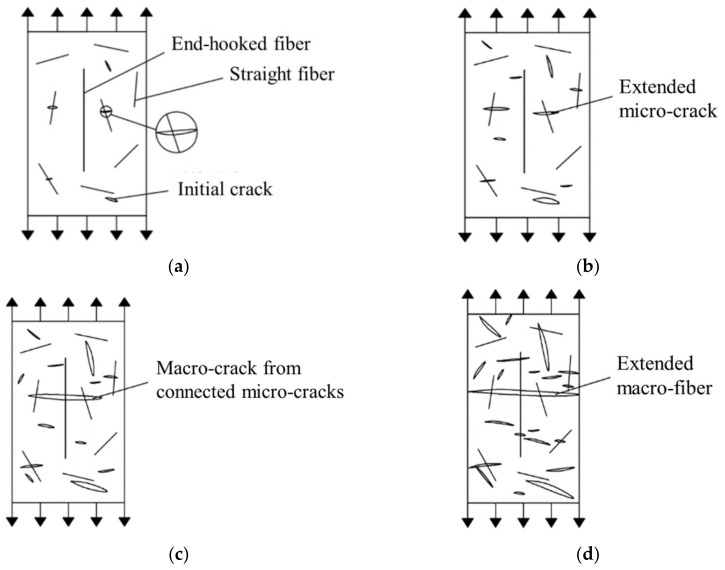
Tensile mechanism figure of hybrid steel fibers. (**a**) Elasticity stage; (**b**) Plasticity stage; (**c**) Peak stage; (**d**) Failure stage.

**Figure 13 materials-16-01114-f013:**
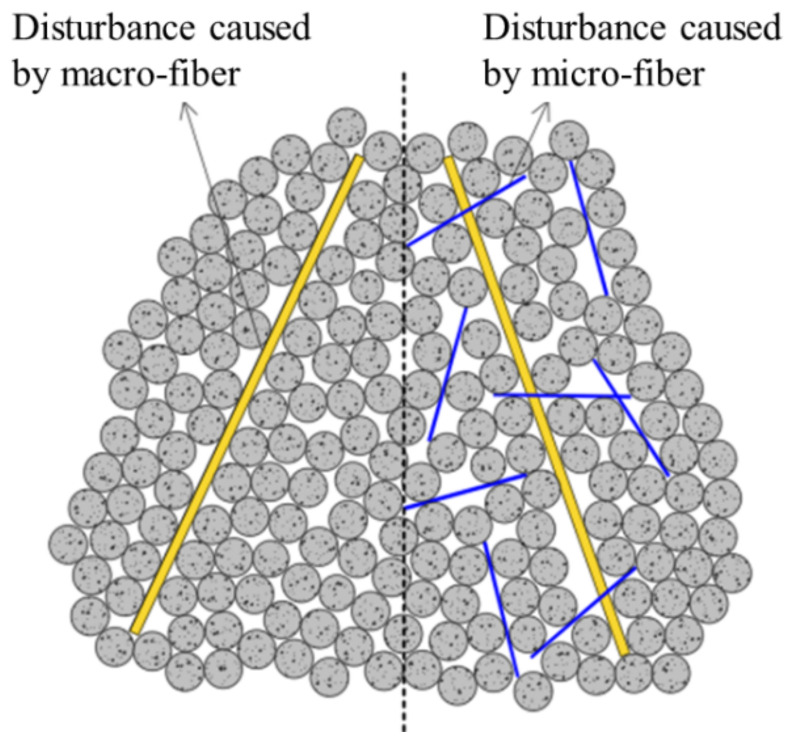
Disturbance effect of steel fiber on aggregate.

**Table 1 materials-16-01114-t001:** Physical parameters of steel fibers.

Fiber Type	Diameter (mm)	Length (mm)	Length to Diameter Ratio	Tensile Strength (MPa)
HS	0.55	30	55	1200
SS	0.20	12	60	2800

**Table 2 materials-16-01114-t002:** Mixture of single or hybrid steel fiber self-compacting concrete.

No.	Mix ID	Amount of Material (kg/m^3^)									
		Water Cement Ratio	Cement	Fly Ash	Slag Powder	Viscosity Modifier	Coarse Aggregate	Fine Sand	Superplasticizer	HS	SS
										Volume Content	
1	H_0_S_0_	0.33	385	62.5	62.5	30	808	808	5–10	0	0
2	H_0_S_2_	0.33	385	62.5	62.5	30	808	808	5–10	0	0.2%
3	H_0_S_4_	0.33	385	62.5	62.5	30	808	808	5–10	0	0.4%
4	H_0_S_6_	0.33	385	62.5	62.5	30	808	808	5–10	0	0.6%
5	H_0_S_8_	0.33	385	62.5	62.5	30	808	808	5–10	0	0.8%
6	H_2_S_0_	0.33	385	62.5	62.5	30	808	808	5–10	0.2%	0
7	H_2_S_2_	0.33	385	62.5	62.5	30	808	808	5–10	0.2%	0.2%
8	H_2_S_4_	0.33	385	62.5	62.5	30	808	808	5–10	0.2%	0.4%
9	H_2_S_6_	0.33	385	62.5	62.5	30	808	808	5–10	0.2%	0.6%
10	H_4_S_0_	0.33	385	62.5	62.5	30	808	808	5–10	0.4%	0
11	H_4_S_2_	0.33	385	62.5	62.5	30	808	808	5–10	0.4%	0.2%
12	H_4_S_4_	0.33	385	62.5	62.5	30	808	808	5–10	0.4%	0.4%
13	H_4_S_6_	0.33	385	62.5	62.5	30	808	808	5–10	0.4%	0.6%
14	H_6_S_0_	0.33	385	62.5	62.5	30	808	808	5–10	0.6%	0
15	H_6_S_2_	0.33	385	62.5	62.5	30	808	808	5–10	0.6%	0.2%
16	H_6_S_4_	0.33	385	62.5	62.5	30	808	808	5–10	0.6%	0.4%
17	H_6_S_6_	0.33	385	62.5	62.5	30	808	808	5–10	0.6%	0.6%
18	H_8_S_0_	0.33	385	62.5	62.5	30	808	808	5–10	0.8%	0

Note: “H” stands for end-hook fiber HS, “S” represents copper-plated microfilament fiber SS, subscript represents content percentage.

**Table 3 materials-16-01114-t003:** Table of CMOD tensile strength parameters.

No.	Mix ID	*f_ctm,fl_* (MPa)	*f_R_*_,1_ (MPa)	*f_R_*_,2_ (MPa)	*f_R_*_,3_ (MPa)	*f_R_*_,4_ (MPa)
1	H_0_S_0_	4.61	0.341	0.039	0.000	0.000
2	H_0_S_2_	5.08	1.354	0.868	0.809	0.670
3	H_0_S_4_	5.09	2.166	1.490	1.148	0.960
4	H_0_S_6_	7.34	5.329	4.238	3.403	2.822
5	H_0_S_8_	8.84	7.358	5.459	4.558	3.820
6	H_2_S_0_	5.82	3.243	3.061	2.050	1.150
7	H_2_S_2_	6.32	4.119	3.610	3.335	2.711
8	H_2_S_4_	5.53	6.153	5.877	4.294	3.573
9	H_2_S_6_	7.76	7.291	5.376	4.266	3.522
10	H_4_S_0_	6.26	5.264	4.861	4.476	3.495
11	H_4_S_2_	7.07	7.065	5.153	4.258	3.646
12	H_4_S_4_	8.68	9.868	7.088	5.264	4.333
13	H_4_S_6_	8.41	9.948	6.531	4.959	4.297
14	H_6_S_0_	6.34	8.209	4.375	3.320	2.918
15	H_6_S_2_	10.80	8.507	6.062	4.701	4.150
16	H_6_S_4_	10.80	12.451	7.701	5.916	4.792
17	H_6_S_6_	10.85	12.227	11.692	9.905	8.317
18	H_8_S_0_	7.00	14.138	13.114	9.395	5.228

## Data Availability

Data are contained within the article.

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
