# Peer review of "Tensile Performance Test Research of Hybrid Steel Fiber—Reinforced Self-Compacting Concrete"

_materials, 2023, doi:10.3390/ma16031114_

Round 1
Reviewer 1 Report
Generally, the content of the paper is quite reasonably and conclusive, but some questions should be clarified.
1. Page 1, line 5: "steel fiber has the potential to replace traditional steel bar": In the opinion of the reviewer this idea seems to be over-reaching. It could be considered in the context of textile reinforced concrete, but in the case of fiber reinforced concrete the weakest link is always present in concrete stuctural elements. After all, the lack of steel reinforcing bars in the beam-column connection could lead to a construction disaster.
2. Page 4, line 1 and 2 from the bottom: Could you clearly add a statement that the load control (the preloading phase) is combined with the displacement control (after that).
3. Page 8 (text) and page 9, Fig. 6: It should be explained directly that not the load-carrying capacity decides about the authors' choice of type of combination of fibes, but the post-peak behaviour. The maximum load for case H8S0 equals almost 18 kN, while for case H6S6 is over 15 kN. Yes, it is written later, but in the opinion of the reviewer is should be clearly discussed on page 8.
Editorial remarks:
1. Page 3, lines 14 and 15: Words "fine", "fineness" and "particle gradation" are repeated. Is it possible to find synonyms (e.g. grind modulus)?
2a. Page 8, line 3 and 2 from the bottom: The slope of the post-peak curve of... instead of: The post-peak curve of...
2b. Please add after that: It means that after the peak the loading rate is the slowest for this case.
3. Page 10, line 2: hsp with the lower index for "sp".
4. References no. 7 and 21 include capital letters.
Author Response
we sincerely thank you for the detailed and constructive comments on our paper. In the attached file, we have concluded the responses to the comments of the three reviewers, in which responses to your comments are also included, so that you can observe our modification to the manuscript.

Reviewer 2 Report
· The informal language is not suitable and should be improved extensively. The article needs major grammatical and syntax improvements. Use of English service center is recommended. Several sentences are not clear and understandable.
· Majority of the qualitative statements should be modified for quantified result comparisons.
· The introduction needs to be revised for higher quality language. The author mentioned some works without stating about the contributions, pros and cons and the how the current work would address.
· The purpose of the article should be clarified in details, why and where this study could be beneficent, more in depth conclusion should be provided.
· The authors mentioned “Concrete is the most widely used building material in the world, but ordinary concrete has some disadvantages such as low tensile strength, low toughness and easy to crack ” The following reference should be added for comprehensiveness of this statement : Experimental investigation of sound transmission loss in concrete containing recycled rubber crumbs.
· The optimized ratio of the fiber and waste should be determined. How the ratios of fiber are chosen for this study?
· All Abbreviations should be expanded.
· More in depth conclusions should be drawn based on various studies, the summary should indicate in depth results and conclusions.
· More descriptive legends and high-quality figures are needed,
· Any figures taken from other works should be reestablished and referenced.
· Figures needs to be professionally done and caption should be more descriptive.
· Equations from other resources should be properly references
Author Response

(The authors gave the same response as above.)

Reviewer 3 Report
The submitted paper materials-2122962 entitled: “Tensile performance test research of hybrid steel fiber reinforced self-compacting concrete” is an experimental study investigating the effects of different fiber content and combination (of hooked ended and straight fibers) on the tensile properties of hybrid steel fiber reinforced concrete (HSFRC). For this purpose, the authors experimentally tested 18 notched beam specimens with different HSFRC mixtures. The authors used digital image correlation (DIC) to obtain the strain field distribution and produce the curves between load and crack mouth opening displacement (CMOD). The manuscript and research presented discusses an interesting and extremely popular field of study however there are some major revisions to be performed before the paper is considered for publication.
1. First of all, the authors have not used the manuscript template of Materials to write their manuscript. The presentation is really poor and it is very hard to follow and comment on the manuscript as there is no line numbering. The references are not properly presented according to mdpi rules.
2. Many of the literature and references used in this manuscript are outdated. This is an extremely popular field of study; the authors should add more references to recent studies in this field along with the necessary explanations in the introduction.
3. Since this is an experimental study, photos of the experimental set-up must be provided and the instrumentation must be explained.
4. Also, providing photos of all tested specimens at least in the final stage of failure is essential.
5. DIC figures of all specimens should also be presented. Providing only figures of three specimens is not enough.
Author Response
We sincerely thank you for the detailed and constructive comments on our paper. In the attached file, we have concluded the responses to the comments of the three reviewers, in which responses to your comments are also included, so that you can observe our modification to the manuscript.

Round 2
Reviewer 1 Report
Editorial corrections are still needed.
Author Response
we sincerely thank you for the detailed and constructive comments on our paper. In the attached file, we have concluded the 2nd round responses to the comments of the three reviewers, in which responses to your comments are also included. In the revised manuscript, you can observe our modification to the manuscript in the revision mode of Microsoft Word.

Reviewer 2 Report
The informal language is not suitable and should be improved extensively. The article needs major grammatical and syntax improvements. Use of English service center is recommended. Several sentences are not clear and understandable.
· The introduction still needs to be revised for higher quality language showing the purpose of the article.
· The authors mentioned “Concrete is the most widely used building material in the world, but ordinary concrete has some disadvantages such as low tensile strength, low toughness and easy to crack ” The following further reference should be added for comprehensiveness of this statement : ) Nano silica and metakaolin effects on the behavior of concrete containing rubber crumbs. CivilEng. ) Investigation of steel fiber effects on concrete abrasion resistance, Advances in concrete construction. ) Compressive behavior of concrete under environmental effects. IntechOpen. ) Temperature and humidity effects on behavior of grouts. Advances in concrete construction.
Author Response

(The authors gave the same response as above.)

Reviewer 3 Report
The authors have partly revised their manuscript but I still insist on presenting photos of all the tested specimens. Since this is an experimental study it is essential to present all the tested specimens, providing them in smaller scale woulf not increase the size of the manuscript.
Also the authors have maybe included new references and deleted some outaded but they just replaced the old with new ones. This is not appropriate. The references must be analysed within the introduction to have clear image of the state of the art.
Author Response

(The authors gave the same response as above.)
